A community perspective on the concept of marine holobionts: current status, challenges, and future directions

Dittami Simon M. simon.dittami@sb-roscoff.fr 1
Arboleda Enrique 2
Auguet Jean-Christophe 3
Bigalke Arite 4
Briand Enora 5
Cárdenas Paco 6
Cardini Ulisse 7
Decelle Johan 8
Engelen Aschwin H. 9
Eveillard Damien 10
Gachon Claire M.M. 11
Griffiths Sarah M. 12
Harder Tilmann 13
Kayal Ehsan 2
Kazamia Elena 14
Lallier François H. 15
Medina Mónica 16
Marzinelli Ezequiel M. 17 18 19
Morganti Teresa Maria 20
Núñez Pons Laura 21
Prado Soizic 22
Pintado José 23
Saha Mahasweta 24 25
Selosse Marc-André 26 27
Skillings Derek 28
Stock Willem 29
Sunagawa Shinichi 30
Toulza Eve 31
Vorobev Alexey 32
Leblanc Catherine leblanc@sb-roscoff.fr 1
Not Fabrice 15
1 Integrative Biology of Marine Models (LBI2M), Station Biologique de Roscoff, Sorbonne Université, CNRS , Roscoff , France
2 FR2424, Station Biologique de Roscoff, Sorbonne Université, CNRS , Roscoff , France
3 MARBEC, Université de Montpellier, CNRS, IFREMER, IRD , Montpellier , France
4 Institute for Inorganic and Analytical Chemistry, Bioorganic Analytics, Friedrich-Schiller-Universität Jena , Jena , Germany
5 Laboratoire Phycotoxines, Ifremer , Nantes , France
6 Pharmacognosy, Department of Medicinal Chemistry, Uppsala University , Uppsala , Sweden
7 Integrative Marine Ecology Dept, Stazione Zoologica Anton Dohrn , Napoli , Italy
8 Laboratoire de Physiologie Cellulaire et Végétale, Université Grenoble Alpes, CNRS, CEA, INRA , Grenoble , France
9 CCMAR, Universidade do Algarve , Faro , Portugal
10 Laboratoire des Sciences Numériques de Nantes (LS2N), Université de Nantes, CNRS , Nantes , France
11 Scottish Marine Institute, Scottish Association for Marine Science , Oban , United Kingdom
12 School of Science and the Environment, Manchester Metropolitan University , Manchester , United Kingdom
13 University of Bremen , Bremen , Germany
14 Institut de Biologie, ENS , Paris , France
15 Adaptation and Diversity in the Marine Environment, Station Biologique de Roscoff, Sorbonne Université, CNRS , Roscoff , France
16 Department of Biology, Pennsylvania State University , University Park , United States of America
17 Ecology and Environment Research Centre, The University of Sydney , Sydney , Australia
18 Singapore Centre for Environmental Life Sciences Engineering, Nanyang Technological University , Singapore , Singapore
19 Sydney Institute of Marine Science , Mosman , Australia
20 Max Planck Institute for Marine Microbiology , Bremen , Germany
21 Section Biology and Evolution of Marine Organisms, Stazione Zoologica Anton Dohrn , Napoli , Italy
22 Molecules of Communication and Adaptation of Microorganisms (UMR 7245), National Museum of Natural History, CNRS , Paris , France
23 Instituto de Investigaciones Marinas, CSIC , Vigo , Spain
24 Benthic Ecology, Helmholtz Center for Ocean Research , Kiel , Germany
25 Marine Ecology and Biodiversity, Plymouth Marine Laboratory , Plymouth , United Kingdom
26 National Museum of Natural History, Département Systématique et Evolution , Paris , France
27 Faculty of Biology, University of Gdansk , Gdansk , Poland
28 Philosophy Department, University of Pennsylvania , Philadelphia , United States of America
29 Laboratory of Protistology & Aquatic Ecology, Department of Biology, Ghent University , Ghent , Belgium
30 Dept. of Biology, Institute of Microbiology and Swiss Institute of Bioinformatics, ETH , Zürich , Switzerland
31 IHPE, Univ. de Montpellier, CNRS, IFREMER, UPDV , Perpignan , France
32 CEA - Institut de Biologie François Jacob, Genoscope , Evry , France
Kormas Konstantinos
Electronic publication date: 2021 Feb 25
Publication date: 2021
Volume: 9
Electronic Location ID: e10911
Received 2020 Oct 6; Accepted 2021 Jan 16
Copyright: ©2021 Dittami et al.
Copyright year: 2021
Copyright holder: Dittami et al.
License: This is an open access article distributed under the terms of the Creative Commons Attribution License, which permits unrestricted use, distribution, reproduction and adaptation in any medium and for any purpose provided that it is properly attributed. For attribution, the original author(s), title, publication source (PeerJ) and either DOI or URL of the article must be cited.
License URL: https://creativecommons.org/licenses/by/4.0/

Keywords: Evolution, Ecosystem services, Symbiosis, Host-microbiota interactions, Marine holobionts, Dysbiosis

Funding: EuroMarine network, Sorbonne University, and the UMRs 8227 and 7144 of the Roscoff Biological Station Australian Research Council Discovery Project DP180104041 The Galician Innovation Agency IN607A 2017/4 The ANR project IDEALG ANR-10-BTBR-04 The European Union’s Horizon 2020 research and innovation program under the Marie Sklodowska-Curie 624575 (ALFF) The ANR project IMPEKAB ANR-15-CE02-001 The Research Council of Lithuania project INBALANCE 09.3.3-LMT-K-712-01-0069 The LabEx GRAL (ANR-10-LABX-49-01) The University of Grenoble Alpes The European Union’s Horizon 2020 research and innovation program through the SponGES project 679849 A Marie Curie Individual Fellowship (Horizon 2020, IRONCOMM) FCT - Foundation for Science and Technology UID/Multi/04326/2019 UIDB/04326/2020 The ATIP-Avenir program The Centre National de Recherche Scientifique This paper is based on the results of a foresight workshop funded by the EuroMarine network, Sorbonne University, and the UMRs 8227 and 7144 of the Roscoff Biological Station. Ezequiel M. Marzinelli was funded by an Australian Research Council Discovery Project (DP180104041), and José Pintado Valverde was funded by the Galician Innovation Agency (IN607A 2017/4). The work of Simon M. Dittami ad Catherine Leblanc was funded by the ANR project IDEALG (ANR-10-BTBR-04). Claire M.M. Gachon, Catherine Leblanc, and Simon M Dittami received funding from the European Union’s Horizon 2020 research and innovation program under the Marie Sklodowska-Curie grant agreement number 624575 (ALFF). The work of Fabrice Not was funded by the ANR project IMPEKAB (ANR-15-CE02-001). Ulisse Cardini was funded by the Research Council of Lithuania project INBALANCE (09.3.3-LMT-K-712-01-0069). Johan Decelle was supported by the CNRS and the ATIP-Avenir program, the LabEx GRAL (ANR-10-LABX-49-01) and Pôle CBS from the University of Grenoble Alpes. Paco Cardenas received support from the European Union’s Horizon 2020 research and innovation program through the SponGES project (grant agreement No. 679849). Elena Kazamia was funded by a Marie Curie Individual Fellowship (Horizon 2020, IRONCOMM). Aschwin H Engelen was supported by Portuguese national funds from FCT - Foundation for Science and Technology through projects UID/Multi/04326/2019 and UIDB/04326/2020. There was no additional external funding received for this study. The funders had no role in study design, data collection and analysis, decision to publish, or preparation of the manuscript.

==============================
Host-microbe interactions play crucial roles in marine ecosystems. However, we still have very little understanding of the mechanisms that govern these relationships, the evolutionary processes that shape them, and their ecological consequences. The holobiont concept is a renewed paradigm in biology that can help to describe and understand these complex systems. It posits that a host and its associated microbiota with which it interacts, form a holobiont, and have to be studied together as a coherent biological and functional unit to understand its biology, ecology, and evolution. Here we discuss critical concepts and opportunities in marine holobiont research and identify key challenges in the field. We highlight the potential economic, sociological, and environmental impacts of the holobiont concept in marine biological, evolutionary, and environmental sciences. Given the connectivity and the unexplored biodiversity specific to marine ecosystems, a deeper understanding of such complex systems requires further technological and conceptual advances, e.g., the development of controlled experimental model systems for holobionts from all major lineages and the modeling of (info)chemical-mediated interactions between organisms. Here we propose that one significant challenge is to bridge cross-disciplinary research on tractable model systems in order to address key ecological and evolutionary questions. This first step is crucial to decipher the main drivers of the dynamics and evolution of holobionts and to account for the holobiont concept in applied areas, such as the conservation, management, and exploitation of marine ecosystems and resources, where practical solutions to predict and mitigate the impact of human activities are more important than ever.

Rationale, intended audience, and survey methodology

The idea of considering organisms in connection with the complex microbial communities they are associated with is a concept rapidly gaining in importance in a wide field of life and environmental sciences. It goes along with an increasing awareness that many organisms depend on complex interactions with their symbiotic microbiota for different aspects of their life, even though the extent of dependencies may vary strongly (Hammer, Sanders & Fierer, 2019). The host and its associated microbiota are considered a single ecological unit, the holobiont. This implies a real paradigm shift. Marine environments harbor most of the diversity of life in terms of the number of lineages that coexist, and the constant presence of surrounding water as a potential carrier of metabolites but also microbes facilitates tight interactions between these lineages, making a “holistic” view of these environments and the organisms that inhabit them particularly important.

This paper is intended for both scientists seeking an overview of recent developments in marine holobiont research and as a reference for policymakers. We review the state of the art in the field research and identify key challenges, possible solutions, and opportunities in the field. Our work is based on the result of a foresight workshop hosted in March 2018, which brought together an interdisciplinary group of 31 scientists. These scientists were selected for their complementary expertise in philosophy, evolution, computer sciences, marine biology, ecology, chemistry, microscopy, and microbiology, as well as for their work with a wide range of different model systems from phytoplankton, via macroalgae, corals, and sponges, to bacterial communities of hydrothermal vents. After a three-day brainstorming session, the resulting ideas and discussions were divided into different topics, and groups of two or more scientists were appointed to draft each section, each based on their expertise in the field, their knowledge of the literature, and literature searches. The assembled paper was then corrected and completed by the entire consortium.

Marine holobionts from their origins to the present

The history of the holobiont concept

Holism is a philosophical notion first proposed by Aristotle in the 4th century BC. It states that systems should be studied in their entirety, with a focus on the interconnections between their various components rather than on the individual parts (Met. Z.17, 1041b11–33). Such systems have emergent properties that result from the behavior of a system that is “larger than the sum of its parts”. However, a major shift away from holism occurred during the Age of “Enlightenment” when the dominant thought summarized as “dissection science” was to focus on the smallest component of a system as a means of understanding it.

The idea of holism started to regain popularity in biology when the endosymbiosis theory was first proposed by Mereschkowski (1905) and further developed by Wallin (1925). Still accepted today, this theory posits a single origin for eukaryotic cells through the symbiotic assimilation of prokaryotes to form first mitochondria and later plastids (the latter through several independent symbiotic events) via phagocytosis (reviewed in Archibald, 2015). These ancestral and founding symbiotic events, which prompted the metabolic and cellular complexity of eukaryotic life, most likely occurred in the ocean (Martin et al., 2008).

Despite the general acceptance of the endosymbiosis theory, the term “holobiosis” or “holobiont” did not immediately enter the scientific vernacular. It was coined independently by Meyer-Abich (1943) (Baedke, Fábregas-Tejeda & Nieves Delgado, 2020) and by Lynn Margulis in 1990, who proposed that evolution has worked mainly through symbiosis-driven leaps that merged organisms into new forms, referred to as “holobionts”, and only secondarily through gradual mutational changes (Margulis & Fester, 1991; O’Malley, 2017). However, the concept was not widely used until it was co-opted by coral biologists over a decade later. Corals and dinoflagellate algae of the family Symbiodiniaceae are one of the most iconic examples of symbioses found in nature; most corals are incapable of long-term survival without the products of photosynthesis provided by their endosymbiotic algae. Rohwer et al. (2002) were the first to use the word “holobiont” to describe a unit of selection sensu Margulis (Rosenberg et al., 2007b) for corals, where the holobiont comprised the cnidarian polyp (host), algae of the family Symbiodiniaceae, various ectosymbionts (endolithic algae, prokaryotes, fungi, other unicellular eukaryotes), and viruses.

Although initially driven by studies of marine organisms, much of the research on the emerging properties and significance of holobionts has since been carried out in other fields of research: the microbiota of the rhizosphere of plants or the animal gut became predominant models and have led to an ongoing paradigm shift in agronomy and medical sciences (Bulgarelli et al., 2013; Shreiner, Kao & Young, 2015; Faure, Simon & Heulin, 2018). Holobionts occur in terrestrial and aquatic habitats alike, and several analogies between these ecosystems can be made. For example, in all of these habitats, interactions within and across holobionts such as induction of chemical defenses, nutrient acquisition, or biofilm formation are mediated by chemical cues and signals in the environment, dubbed infochemicals (Loh et al., 2002; Harder et al., 2012; Rolland et al., 2016; Saha et al., 2019). Nevertheless, we can identify two major differences between terrestrial and aquatic systems. First, the physicochemical properties of water result in higher chemical connectivity and signaling between macro- and micro-organisms in aquatic or moist environments. In marine ecosystems, carbon fluxes also appear to be swifter and trophic modes more flexible, leading to higher plasticity of functional interactions across holobionts (Mitra et al., 2013). Moreover, dispersal barriers are usually lower, allowing for faster microbial community shifts in marine holobionts (Kinlan & Gaines, 2003; Burgess et al., 2016; Martin-Platero et al., 2018). Secondly, phylogenetic diversity at broad taxonomic scales (i.e., supra-kingdom, kingdom and phylum levels), is higher in aquatic realms compared to land, with much of the aquatic diversity yet to be uncovered (De Vargas et al., 2015; Thompson et al., 2017), especially marine viruses (Middelboe & Brussaard, 2017; Gregory et al., 2019). The recent discovery of such astonishing marine microbial diversity in parallel with the scarcity of marine holobiont research suggest a high potential for complex cross-lineage interactions yet to be explored in marine holobionts (Fig. 1).

Figure 1 Partners forming marine holobionts.

They are widespread across the tree of life including all kingdoms (eukaryotes, bacteria, archaea, viruses), and represent a large diversity of potential models for exploring complex biotic interactions across lineages. Plain lines correspond to holobionts referred to in the present manuscript. Dashed lines are examples of potential interactions. Photo credits: Archaeplastida –Catherine Leblanc, Ulisse Cardini; Excavata - Roscoff Culture Collection (http://roscoff-culture-collection.org/rcc-strain-details/1065), Attribution 4.0 International (CC BY 4.0); Amoebozoa - Roscoff Culture Collection (http://roscoff-culture-collection.org/rcc-strain-details/1067), Attribution 4.0 International (CC BY 4.0); Cryptophyta –Roscoff Culture Collection (http://roscoff-culture-collection.org/rcc-strain-details/1998), Attribution 4.0 International (CC BY 4.0); Stramenopila –Catherine Leblanc, Simon M Dittami;Alveolata –Allison Lewis (https://commons.wikimedia.org/wiki/File:Symbiodinium.png), Creative Commons Attribution-Share Alike 4.0 International license; Rhizaria –Fabrice Not; Haptophyta –Alison R. Taylor (https://en.wikipedia.org/wiki/Emiliania_huxleyi#/media/Datei:Emiliania_huxleyi_coccolithophore_(PLoS).png), Attribution 2.5 Generic (CC BY 2.5); Opisthonkonta –HeikeM (https://fr.wikipedia.org/wiki/R%C3%A9cif_corallien_d%27eau_froide#/media/Fichier:Joon1.jpg, Public Domain), NOAA Photo Library (https://en.wikipedia.org/wiki/Sea_anemone#/media/File:Actinoscyphia_aurelia_1.jpg, Public Domain), Squid (Chris Frazee, Margaret McFall-Ngai, https://en.wikipedia.org/wiki/Squid#/media/File:Euprymna_scolopes_-_image.pbio.v12.i02.g001.png, Attribution 4.0 International (CC BY 4.0)); Bacteria –Marinobacter (Astrid Gärdes, Eva Kaeppel, Aamir Shehzad, Shalin Seebah, Hanno Teeling, Pablo Yarza, Frank Oliver Glöckner, Hans-Peter Grossart, Matthias S. Ullrich, https://www.ncbi.nlm.nih.gov/pmc/articles/PMC3035377/figure/f1/, Attribution 2.5 Generic (CC BY 2.5)), Synecococcus (Masur, https://en.wikipedia.org/wiki/Synechococcus#/media/Datei:Synechococcus_PCC_7002_BF.jpg, Public Domain), Vibrio fischeri (Alan Cann, https://www.flickr.com/photos/ajc1/252308050/, Attribution-NonCommercial 2.0 Generic (CC BY-NC 2.0)), Hyphomonas - Holomarine consortium (Simon M Dittami);Archaea –Halobacterium (NASA, https://commons.wikimedia.org/wiki/File:Halobacteria.jpg, Public Domain), Sulfolobus (Xiangyux, https://de.wikipedia.org/wiki/Archaeen#/media/Datei:RT8-4.jpg, Public Domain);Viruses –Matthew B Sullivan, Maureen L Coleman, Peter Weigele, Forest Rohwer, Sallie W Chisholm (https://en.wikipedia.org/wiki/Cyanophage#/media/File:Cyanophages.png), Attribution 2.5 Generic (CC BY 2.5).

The boundaries of holobionts are usually delimited by a physical gradient, which corresponds to the area of local influence of the host, e.g., in unicellular algae the so-called phycosphere (Seymour et al., 2017). However, they may also be defined in a context-dependent way as a “Russian Matryoshka doll”, setting the boundaries of the holobiont depending on the interactions and biological functions that are being considered. Thus holobionts may encompass all levels of host-symbiont associations from intimate endosymbiosis with a high degree of co-evolution up to the community and ecosystem level; a concept referred to as “nested ecosystems” (Fig. 2; McFall-Ngai et al., 2013; Pita et al., 2018).

Figure 2 Schematic view of the “Russian Doll” complexity and dynamics of holobionts, according to diverse spatiotemporal scales.

The host (blue circles), and associated microbes (all other shapes) including bacteria and eukaryotes that may be inside (i.e., endosymbiotic) or outside the host (i.e., ectosymbiotic) are connected by either beneficial (solid orange lines), neutral (solid blue lines) or pathogenic (dashed black lines) interactions, respectively. Changes from beneficial or neutral to pathogenic interactions are typical cases of dysbiosis. The different clusters can be illustrated by the following examples: 1, a model holobiont in a stable physiological condition (e.g., in controlled laboratory condition); 2 and 3, holobionts changing during their life cycle or subjected to stress conditions—examples of vertically transmitted microbes are indicated by light blue arrows; 4 and 5, marine holobionts in the context of global sampling campaigns or long-term time series—examples of horizontal transmission of microbes and holobionts are illustrated by pink arrows.

Such a conceptual perspective raises fundamental questions not only regarding the interaction between the different components of holobionts and processes governing their dynamics, but also of the relevant units of selection and the role of co-evolution. For instance, plant and animal evolution involves new functions co-constructed by members of the holobiont or elimination of functions redundant among them (Selosse, Bessis & Pozo, 2014), and it is likely that these processes are also relevant in marine holobionts. Rosenberg et al. (2010) and Rosenberg & Zilber-Rosenberg (2018) argued that all animals and plants can be considered holobionts, and thus advocate the hologenome theory of evolution, suggesting that natural selection acts at the level of the holobiont and its hologenome. This interpretation of Margulis’ definition of a “holobiont” considerably broadened fundamental concepts in evolution and speciation and has not been free of criticism (Douglas & Werren, 2016), especially when applied at the community or ecosystem level (Moran & Sloan, 2015). More recently, it has been shown that species that interact indirectly with the host can also be important in shaping coevolution within mutualistic multi-partner assemblages (Guimarães et al., 2017). Thus, the holobiont concept and the underlying complexity of holobiont systems should be better defined and further considered when addressing evolutionary and ecological questions.

Marine holobiont models

Today, an increasing number of marine model organisms, both unicellular and multicellular, are being used in holobiont research (Fig. 1), often with different emphasis and levels of experimental control, but altogether covering a large range of scientific topics. Here, we provide several illustrative examples of this diversity and some of the insights they have provided, distinguishing between “environmental models”, chosen for their environmental, evolutionary, economical, or ecological importance, or for historical reasons, but in which microbiome composition is not or only partially controlled, and “controlled bi- or trilateral associations”, which can be kept separately from their symbionts under laboratory conditions and are particularly useful to develop functional approaches and study the mechanisms of symbiotic interactions.

Environmental models: Within the animal kingdom, and in addition to corals and sponges, which will be discussed below, the discovery of deep-sea hydrothermal vents revealed symbioses of animals with chemosynthetic bacteria that have later been found in many other marine ecosystems (Dubilier, Bergin & Lott, 2008; Rubin-Blum et al., 2019) and frequently exhibit high levels of metabolic and taxonomic diversity (Duperron et al., 2008; Petersen et al., 2016; Ponnudurai et al., 2017). In the SAR supergroup, in addition to well-known models such as diatoms, radiolarians and foraminiferans, both heterotrophic protist dwellers harboring endosymbiotic microalgae, are emerging as ecological models for unicellular photosymbiosis due to their ubiquitous presence in the world’s oceans (Decelle, Colin & Foster, 2015; Not et al., 2016). Among the haptophytes, the cosmopolitan Emiliania huxleyi, promoted by associated bacteria (Seyedsayamdost et al., 2011; Segev et al., 2016), produces key intermediates in the carbon and sulfur biogeochemical cycles, making it an important model phytoplankton species. Finally, within the Archaeplastida, the siphonous green alga Bryopsis is an example of a model that harbors heterotrophic endosymbiotic bacteria, some of which exhibit patterns of co-evolution with their hosts (Hollants et al., 2013).

Controlled bi- or trilateral associations: Only a few models, covering a small part of the overall marine biodiversity, are currently being cultivated ex-situ and can be used in fully controlled experiments, where they can be cultured aposymbiotically. The flatworm Symsagittifera (= Convoluta) roscoffensis (Arboleda et al., 2018), the sea anemone Exaiptasia (Baumgarten et al., 2015; Wolfowicz et al., 2016), the upside-down jellyfish Cassiopea (Ohdera et al., 2018), and their respective intracellular green and dinoflagellate algae have, in addition to corals, become models for fundamental research on evolution of metazoan-algal photosymbiosis. In particular, the sea anemone Exaiptasia has been used to explore photobiology disruption and restoration of cnidarian symbioses (Lehnert, Burriesci & Pringle, 2012). The Vibrio-squid model provides insights into the effect of microbiota on animal development, circadian rhythms, and immune systems (McFall-Ngai, 2014). The unicellular green alga Ostreococcus, an important marine primary producer, has been shown to exchange vitamins with specific associated bacteria (Cooper et al., 2019). The green macroalga Ulva mutabilis has enabled the exploration of bacteria-mediated growth and morphogenesis including the identification of original chemical interactions in the holobiont (Wichard, 2015; Kessler et al., 2018). Although the culture conditions in these highly-controlled model systems differ from the natural environment, these systems are essential to gain elementary mechanistic understanding of the functioning, the roles, and the evolution of marine holobionts.

The influence of marine holobionts on ecological processes

Work on model systems has demonstrated that motile and macroscopic marine holobionts can act as dissemination vectors for geographically restricted microbial taxa. Pelagic mollusks or vertebrates are textbook examples of high dispersal capacity organisms (e.g., against currents and through stratified water layers). It has been estimated that fish and marine mammals may enhance the original dispersion rate of their microbiota by a factor of 200 to 200,000 (Troussellier et al., 2017) and marine birds may even act as bio-vectors across ecosystem boundaries (Bouchard Marmen et al., 2017). This host-driven dispersal of microbes can include non-native or invasive species as well as pathogens (Troussellier et al., 2017).

A related ecological function of holobionts is their potential to sustain rare species. Hosts provide an environment that favors the growth of specific microbial communities distinct from the surrounding environment (including rare microbes). They may, for instance, provide a nutrient-rich niche in the otherwise nutrient-poor surroundings (Smriga, Sandin & Azam, 2010; Webster et al., 2010; Burke et al., 2011a; Burke et al., 2011b; Chiarello et al., 2018).

Lastly, biological processes regulated by microbes are important drivers of global biogeochemical cycles (Falkowski, Fenchel & Delong, 2008; Madsen, 2011; Anantharaman et al., 2016). In the open ocean, it is estimated that symbioses with the cyanobacterium UCYN-A contribute ∼20% to total N2 fixation (Thompson et al., 2012; Martínez-Pérez et al., 2016). In benthic systems, sponges and corals may support entire ecosystems via their involvement in nutrient cycling thanks to their microbial partners (Raina et al., 2009; Fiore et al., 2010; Cardini et al., 2015; Pita et al., 2018), functioning as sinks and sources of nutrients. In particular the “sponge loop” recycles dissolved organic matter and makes it available to higher trophic levels in the form of detritus (De Goeij et al., 2013; Fiore et al., 2010; Rix et al., 2017). In coastal sediments, bivalves hosting methanogenic archaea have been shown to increase the benthic methane efflux by a factor of up to eight, potentially accounting for 9.5% of total methane emissions from the Baltic Sea (Bonaglia et al., 2017). Such impressive metabolic versatility is accomplished because of the simultaneous occurrence of disparate biochemical machineries (e.g., aerobic and anaerobic pathways) in individual symbionts, providing new metabolic abilities to the holobiont, such as the synthesis of specific essential amino acids, photosynthesis, or chemosynthesis (Dubilier, Bergin & Lott, 2008; Venn, Loram & Douglas, 2008). Furthermore, the interaction between host and microbiota can potentially extend the metabolic capabilities of a holobiont in a way that augments its resilience to environmental changes (Berkelmans & Van Oppen, 2006; Gilbert et al., 2010; Dittami et al., 2016; Shapira, 2016; Godoy et al., 2018), or allow it to cross biotope boundaries (e.g., Woyke et al., 2006) and colonize extreme environments (Bang et al., 2018). Holobionts thus contribute to marine microbial diversity and possibly resilience in the context of global environmental changes (Troussellier et al., 2017) and it is paramount to include the holobiont concept in predictive models that investigate the consequences of human impacts on the marine realm and its biogeochemical cycles.

Challenges and opportunities in marine holobiont research

Marine holobiont assembly and regulation

Two critical challenges partially addressed by using model systems are (1) to decipher the factors determining holobiont composition and (2) to elucidate the impacts and roles of the different partners in these complex systems over time. Some marine organisms such as bivalves transmit part of the microbiota maternally (Bright & Bulgheresi, 2010; Funkhouser & Bordenstein, 2013). In other marine holobionts, vertical transmission may be weak and inconsistent, whereas mixed modes of transmission (vertical and horizontal) or intermediate modes (pseudo-vertical, where horizontal acquisition frequently involves symbionts of parental origin) are more common (Björk et al., 2019). Identifying the factors shaping holobiont composition and understanding their evolution is highly relevant for marine organisms given that most marine hosts display a high specificity for their microbiota and even patterns of phylosymbiosis (Brooks et al., 2016; Kazamia et al., 2016; Pollock et al., 2018), despite a highly connected and microbe-rich environment.

During microbiota transmission (whether vertical or horizontal), selection by the host and/or by other components of the microbiome, is a key process in establishing or maintaining a holobiont microbial community that is distinct from the environment. The immune system of the host, e.g., via the secretion of specific antimicrobial peptides (Franzenburg et al., 2013; Zheng, Liwinski & Elinav, 2020), is one way of performing this selection in both marine and terrestrial holobionts.

Another way of selecting a holobiont microbial community is by chemically mediated microbial gardening. This concept has been demonstrated for land plants, where root exudates manipulate microbiome composition (Lebeis et al., 2015). In marine environments, the phylogenetic diversity of hosts and symbionts suggests both conserved and marine-specific chemical interactions, but studies are still in their infancy. For instance, seaweeds can chemically garden beneficial microbes, facilitating normal morphogenesis and increasing disease resistance (Kessler et al., 2018; Saha & Weinberger, 2019), and seaweeds and corals structure their surface-associated microbiome by producing chemo-attractants and anti-bacterial compounds (Harder et al., 2012; Ochsenkühn et al., 2018). There are fewer examples of chemical gardening in unicellular hosts, but it seems highly likely that similar processes are in place (Gribben et al., 2017; Cirri & Pohnert, 2019).

In addition to selection, ecological drift, dispersal and evolutionary diversification have been proposed as key processes in community assembly, but are difficult to estimate in microbial communities (Nemergut et al., 2013). The only data currently at our disposal to quantify these processes are the diversity and distribution of microbes. Considering the high connectivity of aquatic environments, differences in marine microbial communities are frequently attributed to a combination of selection and drift, rather than limited dispersal (e.g., Burke et al., 2011a), a conclusion which, in the future, could be refined by conceptual models developed for instance for soil microbial communities (Stegen et al., 2013; Dini-Andreote et al., 2015). Diversification is mainly considered in the sense of coevolution or adaptation to host selection, which may also be driven by the horizontal acquisition of genes. However, cospeciation is challenging to prove (De Vienne et al., 2013; Moran & Sloan, 2015) and only few studies have examined this process in marine holobionts to date, each focused on a restricted number of actors (e.g., Peek et al., 1998; Lanterbecq, Rouse & Eeckhaut, 2010).

Perturbations in the transmission or the recruitment of the microbiota can lead to dysbiosis, and eventually microbial infections (Selosse, Bessis & Pozo, 2014; De Lorgeril et al., 2018). Dysbiotic microbial communities are frequently determined by stochastic processes and thus display higher variability in their composition than those of healthy individuals. This observation in line with the “Anna Karenina principle” (Zaneveld, McMinds & Vega Thurber, 2017), although there are exceptions to this rule (e.g., Marzinelli et al., 2015). A specific case of dysbiosis is the so-called “Rasputin effect” where benign endosymbionts opportunistically become detrimental to the host due to processes such as reduction in immune response under food deprivation, coinfections, or environmental pressure (Overstreet & Lotz, 2016). Many diseases are now interpreted as the result of a microbial imbalance and the rise of opportunistic or polymicrobial infections upon host stress (Egan & Gardiner, 2016). For instance in reef-building corals, warming destabilizes cnidarian-dinoflagellate associations, and some beneficial Symbiodiniacea strains switch their physiology and sequester more resources for their own growth at the expense of the coral host, leading to coral bleaching and even death (Baker et al., 2018).

Increasing our knowledge on the contribution of these processes to holobiont community assembly in marine systems is a key challenge, which is of particular urgency today in the context of ongoing global change. Moreover, understanding how the community and functional structure of resident microbes are resilient to perturbations remains critical to predict and promote the health of their host and the ecosystem. Yet, the contribution of the microbiome is still missing in most quantitative models predicting the distribution of marine macro-organisms, or additional information on biological interactions would be required to make the former more accurate (Bell et al., 2018).

Integrating marine model systems with large-scale studies

By compiling a survey of the most important trends and challenges in the field of marine holobiont research (Fig. 3), we identified two distinct opinion clusters: one focused on mechanistic understanding and work with model systems whereas another targets large-scale and heterogeneous data set analyses and predictive modeling. This illustrates that, on the one hand, the scientific community is interested in the establishment of models for the identification of specific molecular interactions between marine organisms at a given point in space and time, up to the point of synthesizing functional mutualistic communities in vitro (Kubo et al., 2013). On the other hand, another part of the community is moving towards global environmental sampling schemes such as the TARA Oceans expedition (Pesant et al., 2015) or the Ocean Sampling Day (Kopf et al., 2015), and towards long-term data series (e.g., Wiltshire et al., 2010; Harris, 2010). What emerges as both lines of research progress is the understanding that small-scale functional studies in the laboratory are inconsequential unless made applicable to ecologically-relevant systems. At the same time, and despite the recent advances in community modeling (Ovaskainen et al., 2017), hypotheses drawn from large scale-studies remain correlative and require experimental validation of the mechanisms driving the observed processes. We illustrate the importance of integrating both approaches in Fig. 3, where the node related to potential applications was perceived as a central hub at the interface between mechanistic understanding and predictive modeling.

Figure 3 Mind map of key concepts, techniques, and challenges related to marine holobionts.

The basis of this map was generated during the Holomarine workshop held in Roscoff in 2018 (https://www.euromarinenetwork.eu/activities/HoloMarine). The size of the nodes reflects the number of votes each keyword received from the participants of the workshop (total of 120 votes from 30 participants). The two main clusters corresponding to predictive modeling and mechanistic modeling, are displayed in purple and turquoise, respectively. Among the intermediate nodes linking these disciplines (blue) “potential use, management” was the most connected.

A successful example merging both functional and large-scale approaches, are the root nodules of legumes, which harbor nitrogen-fixing bacteria. In this system, the functioning, distribution, and to some extent the evolution of these nodules, are now well understood (Epihov et al., 2017). The integration of this knowledge into agricultural practices has led to substantial yield improvements (e.g., Kavimandan, 1985; Alam et al., 2015). In the more diffuse and partner-rich system of mycorrhizal symbioses between plant roots and soil fungi, a better understanding of the interactions has also been achieved via the investigation of environmental diversity patterns in combination with experimental culture systems with reduced diversity (Van der Heijden et al., 2015).

We advocate the implementation of comparable efforts in marine sciences through interdisciplinary research combining physiology, biochemistry, ecology, and computational modeling. A key factor will be the identification and development of tractable model systems for keystone holobionts that allow hypotheses generated by large-scale data sets to be tested in controlled experiments. Such approaches will enable the identification of organismal interaction patterns within holobionts and nested ecosystems. In addition to answering fundamental questions, they will help address the ecological, societal, and ethical issues that arise from attempting to actively manipulate holobionts (e.g., in aquaculture, conservation, and invasion) in order to enhance their resilience and protect them from the impacts of global change (Llewellyn et al., 2014).

Emerging methodologies to approach the complexity of holobiont partnerships

As our conceptual understanding of the different levels of holobiont organization evolves, so does the need for multidisciplinary approaches and the development of tools and technologies to handle the unprecedented amount of data and their integration into dedicated ecological and evolutionary models. Here, progress is often fast-paced and provides exciting opportunities to address some of the challenges in holobiont research.

A giant technological stride has been the explosion of affordable “omics” technologies allowing molecular ecologists to move from metabarcoding (i.e., sequencing of a taxonomic marker) to metagenomics or single-cell genomics, metatranscriptomics, and metaproteomics, thus advancing our research from phylogenetic analyses of the holobiont to analyses capable of making predictions about the functions carried out by different components of the holobiont (Bowers, Doud & Woyke, 2017; Meng et al., 2018; Fig. 4). These approaches are equally useful in marine and in terrestrial environments, but the scarcity of well-studied lineages in the former makes the generation of good annotations and reference databases challenging for marine biologists. Metaproteomics combined with stable isotope fingerprinting can help study the metabolism of single lineages within the holobiont (Kleiner et al., 2018). In parallel, meta-metabolomics approaches have advanced over the last decades, and can be used to unravel the chemical interactions between partners. One limitation particularly relevant to marine systems is that many compounds are often not referenced in the mostly terrestrial-based databases, although recent technological advances such as molecular networking and meta-mass shift chemical profiling to identify relatives of known molecules may help to overcome this challenge (Hartmann et al., 2017).

Figure 4 Impact of emerging methodologies (light green) on the main challenges in marine holobiont research identified in this paper (blue).

Turquoise and purple correspond to the two main clusters of activity identified in Fig. 3.

A further challenge in holobiont research is to identify the origin of compounds among the different partners of the holobionts and to determine their involvement in the maintenance and performance of the holobiont system. Well-designed experimental setups may help answer some of these questions (e.g., Quinn et al., 2016), but they will also require high levels of replication in order to represent the extensive intra-species variability found in marine systems. Recently developed in vivo and in situ imaging techniques combined with metabolomicomics can provide small-scale spatial and qualitative information (origin, distribution, and concentration of a molecule or nutrient), shedding new light on the contribution of each partner of the holobiont system at the molecular level (e.g., Geier et al., 2020). The combination of stable isotope labelling and chemical imaging (mass spectrometry imaging such as secondary ion mass spectrometry and matrix-assisted laser desorption ionization, and synchrotron X-ray fluorescence) is particularly valuable in this context, as it enables the investigation of metabolic exchange between the different compartments of a holobiont (Musat et al., 2016; Raina et al., 2017). Finally, three-dimensional electron microscopy may help evaluate to what extent different components of a holobiont are physically integrated (Colin et al., 2017; Decelle et al., 2019), where high integration is one indication of highly specific interactions. All of these techniques can be employed in both marine and terrestrial systems, but in marine systems the high phylogenetic diversity of organisms adds to the complexity of adapting and optimizing these techniques.

One consequence of the development of such new methods is the feedback they provide to improve existing models or to develop entirely new ones, e.g., by conceptualizing holobionts as the combination of the interactions between the host and its microbiota (Skillings, 2016; Berry & Loy, 2018), or by redefining boundaries between the holobiont and its environment (Zengler & Palsson, 2012). Such models may incorporate metabolic complementarity between different components of the holobiont (Dittami, Eveillard & Tonon, 2014; Bordron et al., 2016), e.g., enabling the prediction of testable metabolic properties depending on holobiont composition (Burgunter-Delamare et al., 2020), or simulate microbial communities starting from different cohorts of randomly generated microbes for comparison with actual metatranscriptomics and/or metagenomics data (Coles et al., 2017).

A side-effect of these recent developments has been to move holobiont research away from laboratory culture-based experiments. We argue that maintaining or even extending cultivation efforts, possibly via the implementation of “culturomics” approaches as successfully carried out for the human gut microbiome (Lagier et al., 2012), remains essential to capture the maximum holobiont biodiversity possible and will facilitate the experimental testing of hypotheses and the investigation of physiological mechanisms. A striking example of the importance of laboratory experimentation is the way germ-free mice re-inoculated with cultivated bacteria (the so-called gnotobiotic mice) have contributed to the understanding of interactions within the holobiont in animal health, physiology, and behavior (e.g., Neufeld et al., 2011; Faith et al., 2014; Selosse, Bessis & Pozo, 2014). In terms of gnotobiotic marine organisms there are several examples of microalgae that can be cultured axenically, but gnotobiotic multicellular organisms are rare. One example is the green alga Ulva mutabilils, which can be rendered axenic based on the movement of its spores and has been used to study the effects of bacteria-produced morphogens (Spoerner et al., 2012). There are also examples of gnotobiotic marine fish and mollusks (Marques et al., 2006). However, in many cases, not all associated microorganisms can be controlled leaving researchers with aposymbiotic cultures (i.e., cultures without the main symbiont(s), as e.g., for the sea anemone Exaiptasia) (Lehnert et al., 2014). Innovations in cultivation techniques for axenic (or germ-free) hosts or in microbial cultivation such as microfluidic systems (e.g., Pan et al., 2011) and cultivation chips (Nichols et al., 2010) may provide a way to obtain a wider spectrum of pure cultures. Yet, bringing individual components of holobionts into cultivation can still be a daunting challenge due to the strong interdependencies between organisms as well as the existence of yet unknown metabolic processes that may have specific requirements. In this context, single-cell “omics” analyses can provide critical information on some of the growth requirements of the organisms, and complement approaches of high-throughput culturing (Gutleben et al., 2018).

Established cultures can then be developed into model systems, e.g., by genome sequencing and the development of genetic tools to move towards mechanistic understanding and experimental testing of hypothetical processes within the holobiont derived from environmental meta “omics” approaches. In this context, CRISPR/cas9 is a particularly promising tool for the genetic modification of both host and symbiont organisms, and has been established for a few marine model systems, including diatoms, cnidarians, annelids, echinoderms, and chordates (Momose & Concordet, 2016), although this tool has not, to the best of our knowledge, been used so far to decipher host symbiont interactions. “Omics” techniques, coupled to efforts in adapting these genetic tools, have the potential to broaden the range of available models, enabling a better understanding of the functioning of marine holobionts and their interactions in marine environments (Wichard & Beemelmanns, 2018).

Ecosystem services and holobionts in natural and managed systems

A better understanding of marine holobionts will likely have direct socio-economic consequences for coastal marine ecosystems, estimated to provide services worth almost 50 trillion (1012) US$ per year (Costanza et al., 2014). Most of the management practices in marine systems have so far been based exclusively on the biology and ecology of macro-organisms. A multidisciplinary approach that provides mechanistic understanding of habitat-forming organisms as holobionts will ultimately improve the predictability and management of coastal ecosystems. For example, host-associated microbiota could be integrated in biomonitoring programs as proxies used to assess the health of ecosystems. Microbial shifts and dysbiosis constitute early warning signals that may allow managers to predict potential impacts and intervene more rapidly and effectively (Van Oppen et al., 2017; Marzinelli et al., 2018).

One form of intervention could be to promote positive changes of host-associated microbiota, in ways analogous to the use of pre- and/or probiotics in humans (Singh et al., 2013) or inoculation of beneficial microbes in plant farming (Berruti et al., 2015; Van der Heijden et al., 2015). In macroalgae, beneficial bacteria identified from healthy seaweed holobionts could be used as biological control agents and applied to diseased plantlets in order to suppress the growth of bacteria detrimental to the host and to prevent disease outbreaks in aquaculture settings. In addition to bacteria, these macroalgae frequently host endophytic fungi that may have protective functions for the algae (Porras-Alfaro & Bayman, 2011; Vallet et al., 2018). Host-associated microbiota could also be manipulated to shape key phenotypes in cultured marine organisms. For example, specific bacteria associated with microalgae may enhance algal growth (Amin et al., 2009; Kazamia, Aldridge & Smith, 2012; Le Chevanton et al., 2013), increase lipid content (Cho et al., 2015), and participate in the bioprocessing of algal biomass (Lenneman, Wang & Barney, 2014). More recently, the active modification of the coral microbiota has even been advocated as a means to boost the resilience of the holobiont to climate change (Van Oppen et al., 2015; Peixoto et al., 2017), an approach which would, however, bear a high risk of unanticipated and unintended side effects.

Also, holistic approaches could be implemented in the framework of fish farms. Recent developments including integrated multi-trophic aquaculture, recirculating aquaculture, offshore aquaculture, species selection, and breeding increase yields and reduce the resource constraints and environmental impacts of intensive aquaculture (Klinger & Naylor, 2012). However, the intensification of aquaculture often goes hand in hand with increased amounts of disease outbreaks both in industry and wild stocks. A holistic microbial management approach, e.g., by reducing the use of sterilization procedures and favoring the selection of healthy and stable microbiota consisting of slow-growing K-strategists, may provide an efficient solution to these latter problems, reducing the sensitivity of host to opportunistic pathogens (De Schryver & Vadstein, 2014).

Nevertheless, when considering their biotechnological potential, it should also be noted that marine microbiota are likely vulnerable to anthropogenic influences and that their deliberate engineering, introduction from exotic regions (often hidden in their hosts), or inadvertent perturbations may have profound, and yet entirely unknown, consequences for marine ecosystems. Terrestrial environments provide numerous examples of unwanted plant expansions or ecosystem perturbations linked to microbiota (e.g., Dickie et al., 2017), and cases where holobionts manipulated by human resulted in pests (e.g., Clay & Holah, 1999) call for a cautious and ecologically-informed evaluation of holobiont-based technologies in marine systems.

Conclusions

Marine ecosystems represent highly connected reservoirs of largely unexplored biodiversity. They are of critical importance to feed the ever-growing world population, constitute significant players in global biogeochemical cycles but are also threatened by human activities and global change. In order to unravel some of the basic principles of life and its evolution, and to protect and sustainably exploit marine natural resources, it is paramount to consider the complex biotic interactions that shape the marine communities and their environment. The scope of these interactions ranges from simple molecular signals between two partners, via complex assemblies of eukaryotes, prokaryotes, and viruses with one or several hosts, to entire ecosystems. Accordingly, current key questions in marine holobiont research cover a wide range of topics: What are the exchanges that occur between different partners of the holobiont, and how do they condition their survival, dynamics and evolution? What are the cues and signals driving these exchanges? What are the relevant units of selection and dispersal in marine holobionts? How do holobiont systems and the interactions within them change over time and in different conditions? How do such changes impact ecological processes? How can this knowledge be applied to our benefit and where do we need to draw limits? Identifying and consolidating key model systems while adapting emerging “-omics”, imaging, culturing technologies, and functional analyses via transgenesis (e.g., CRISPR/cas9) to them will be critical to the development of “holobiont-aware” ecosystem models.

The concept of holobionts represents the fundamental understanding that all living organisms have intimate connections with their immediate neighbors, which may impact all aspects of their biology. We believe that this concept of holobionts will be most useful if used with a degree of malleability, enabling us to define units of interacting organisms that are most suitable to answer specific questions. The consideration of the holobiont concept marks a paradigm shift in biological and environmental sciences, but only if scientists work together as an (inter)active and transdisciplinary community bringing together holistic and mechanistic views. This will result in tangible outcomes including a better understanding of evolutionary and adaptive processes, improved modeling of habitats and understanding of biogeochemical cycles, as well as application of the holobiont concept in aquaculture and ecosystem management projects.

This document reflects only the authors’ view and the Executive Agency for Small and Medium-sized Enterprises (EASME) is not responsible for any use that may be made of the information it contains. This manuscript has been peer-reviewed and recommended by Peer Community In Ecology (https://doi.org/10.24072/pci.ecology.100045).

Glossary

* If no other examples of the use of each term are cited below, the definition was based on the online version of the Merriam-Webster dictionary (2019, https://www.merriam-webster.com/) or the Oxford dictionary (2020, https://www.lexico.com/)

Anna Karenina principle several factors can cause a system to fail, but only a narrow range of parameters characterizes a working system; based on the first sentence of Leo Tolstoy’s “Anna Karenina” (1878): “Happy families are all alike; every unhappy family is unhappy in its own way” (Zaneveld, McMinds & Vega Thurber, 2017)

Aposymbiotic culture a culture of a host or a symbiont without its main symbiotic partner(s) (e.g., Kelty & Cook, 1976). In contrast to gnotobiotic cultures, aposymbiotic cultures are usually not germ-free

Biological control (biocontrol) methods for controlling diseases or pests by introducing or supporting natural enemies of the former (see e.g., Hoitink & Boehm, 1999)

Biomonitoring the use of living organisms as quantitative indicator for the health of an environment or ecosystem (Holt & Miller, 2010)

Community assembly process the processes that shape community composition in a given habitat, according to Nemergut et al. (2013) the four main forces relevant for community assembly are evolutionary diversification, dispersal, selection, and ecological drift

Dysbiosis microbial imbalance in a symbiotic community that affects the health of the host (Egan & Gardiner, 2016)

Ecological process the processes responsible for the functioning and dynamics of ecosystems including biogeochemical cycles, community assembly processes, interactions between organisms, and climatic processes (see e.g., Bennett et al., 2009)

Ecosystem services any direct or indirect benefits that humans can draw from an ecosystem; they include provisioning services (e.g., food), regulating services (e.g., climate), cultural services (e.g., recreation), and supporting services (e.g., habitat formation) (Millennium Ecosystem Assessment Panel, 2005)

Ectosymbiosis a symbiotic relationship in which symbionts live on the surface of a host. This includes, for instance, algal biofilms or the skin microbiome (Nardon & Charles, 2001)

Emergent property a property of complex systems (e.g., holobionts), which arises from interactions between the components and that is not the sum of the component properties (see e.g., Theis, 2018)

Endosymbiosis (sometimes also referred to more precisely as endocytobiosis; Nardon & Charles, 2001) –a symbiotic relationship in which a symbiont lives inside the host cells; prominent examples are mitochondria, plastids/photosymbionts, or nitrogen fixing bacteria in plant root nodules. See also ectosymbioisis

Gnotobiosis the condition in which all organisms present in a culture can be controlled, i.e., germ-free (axenic) organisms or organisms with a controlled community of symbionts. Gnotobiotic individuals may be obtained e.g., by surgical removal from the mother (vertebrates) or by surface sterilization of seeds (plants) and subsequent handling in a sterile environment and possible inoculation with selected microbes (Hale, Lindsey & Hameed, 1973; Williams, 2014)

Holism the theory that parts of a whole are in intimate interconnection, such that they cannot exist independently of the whole, or cannot be understood without reference to the whole, which is thus regarded as greater than the sum of its parts

Holobiont an ecological unit of different species living together in symbiosis. The term is frequently used for the unit of a host and its associated microbiota but can be extended to larger scales. Whether or to what extent holobionts are also a unit of evolution is still a matter of debate (Douglas & Werren, 2016)

Hologenome the combined genomes of the host and all members of its microbiota; (Rosenberg et al., 2007a; Zilber-Rosenberg & Rosenberg, 2008)

Horizontal transmission acquisition of the associated microbiome from the environment (e.g., Myers & Rothman, 1995; Roughgarden, 2019)

Host the largest or dominant partner in a holobiont

Infochemical a chemical compound, usually diffusible, that carries information on the environment, such as the presence of other organisms, and can be used to mediate inter- and intraspecific communication (Dicke & Sabelis, 1988)

Microbial gardening the act of frequently releasing growth-enhancing or inhibiting chemicals or metabolites that favor the development of a microbial community beneficial to the host (see e.g., Saha & Weinberger, 2019)

Microbiome the combined genetic information encoded by the microbiota; may also refer to the microbiota itself or the microbiota and its environment (see Marchesi & Ravel, 2015)

Microbiota all microorganisms present in a particular environment or associated with a particular host (see Marchesi & Ravel, 2015)

Nested ecosystems a view of ecosystems where each individual system, like a “Russian doll”, can be decomposed into smaller systems and/or considered part of a larger system, all of which still qualify as ecosystems (e.g., McFall-Ngai et al., 2013)

Phagocytosis a process by which a eukaryotic cell ingests other cells or solid particles, e.g., the engulfing of symbionts by sponges (Leys et al., 2018)

Phycosphere the physical envelope surrounding a phytoplankton cell; usually rich in organic matter (see Amin, Parker & Armbrust, 2012)

Phylosymbiosis congruence in the phylogeny of different hosts and the composition of their associated microbiota (Brooks et al., 2016)

Rasputin effect the phenomenon that commensals and mutualists can become parasitic in certain conditions (Overstreet & Lotz, 2016); after the Russian monk Rasputin who became the confidant of the Tsar of Russia, but later helped bring down the Tsar’s empire during the Russian revolution

Sponge loop sponges efficiently recycle dissolved organic matter turning it into detritus that becomes food for other consumers (De Goeij et al., 2013)

Symbiont an organism living in symbiosis; usually refers to the smaller/microbial partners living in mutualistic relationships (see also host), but also includes organisms in commensalistic and parasitic relationships

Symbiosis a close and lasting or recurrent (e.g., over generations) relationship between organisms living together; usually refers to mutualistic relationships, but also includes commensalism and parasitism

Vertical transmission acquisition of the associated microbiome by a new generation of hosts from the parents (as opposed to horizontal transmission; e.g., Roughgarden, 2019)

Additional Information and Declarations

Competing Interests

Author Contributions

Data Availability

Fabrice Not is currently a PeerJ Academic Editor. Monica Medina is a former PeerJ Academic Editor.

Simon M. Dittami, Willem Stock, Catherine Leblanc and Fabrice Not conceived and designed the experiments, performed the experiments, analyzed the data, prepared figures and/or tables, authored or reviewed drafts of the paper, and approved the final draft.

Enrique Arboleda, Jean-Christophe Auguet, Arite Bigalke, Enora Briand, Paco Cárdenas, Ulisse Cardini, Johan Decelle, Aschwin H. Engelen, Damien Eveillard, Claire M.M. Gachon, Sarah M. Griffiths, Tilmann Harder, Ehsan Kayal, Elena Kazamia, François H. Lallier, Mónica Medina, Ezequiel M. Marzinelli, Teresa Maria Morganti, Laura Núñez Pons, Soizic Prado, José Pintado, Mahasweta Saha, Marc-André Selosse, Derek Skillings, Shinichi Sunagawa, Eve Toulza and Alexey Vorobev conceived and designed the experiments, performed the experiments, analyzed the data, authored or reviewed drafts of the paper, and approved the final draft.

The following information was supplied regarding data availability:

This is a review paper and does not have associated raw data or code.

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
