# Peer review of "A community perspective on the concept of marine holobionts: current status, challenges, and future directions"

_PeerJ, doi:10.7717/peerj.10911_

## Round 0.1 · original submission · Minor Revisions

Please provide a point-by-point response to all of the reviewers' comments.

·

Basic reporting

Dittami et al. is a product of The HoloMarine working group, a collection of experts with diverse backgrounds and a common interest in understanding how eukaryote-symbiont relationships influence the marine ecosystem. Through a multi-day workshop, this group set out to “discuss critical concepts and opportunities in marine holobiont research and identify key challenges in the field” by highlighting “the potential economic, sociological, and environmental impacts of the holobiont concept in marine biological, evolutionary, and environmental sciences.” This is a bold endeavor. Like many others before it, these exercises are inevitably designed (with rare exception; e.g., McFall-Ngai et al., 2013, PNAS) to manifest into a reiteration of similar communal activities (e.g., Wilkins et al., 2019, PLoS Biology). This manuscript followed that tune by reciting major concepts in holobiont research using common examples and by not providing a novel, forward-thinking synthesis that may serve as a backbone for future research. However, it is only right to attest that the criticisms above reflect personal standards for working groups and this should not hold back or delay this manuscript from publication but more of serve as food-for-though in telling this story. Dittami et al. did provide a well-written manuscript piece that hits on major topics in holobiont research and could serve as a nice resource for those interested in the topic. Provided the manuscript at hand and the revisions it has gone through, I have a few comments that I wish for them to be addressed as part of a minor revision.


First, “The history of the holobiont concept” comes off from the lone perspective that the microbiome is that last missing biological link. While there are components of this that may be true, it does not reflect how the holobiont concept is a spectrum, from some species being heavily reliant on symbionts to barely, if at all. This concept, that was brilliantly addressed by Hammer et al. (2019, FEMS Microbiology Letters) and should be reflected in this, as well.

Second, “Emerging methodologies to approach the complexity of holobiont partnerships” provides the perspective that -omics is the way of the future for this type of research. Recent trends suggest otherwise. While sequencing will, without question, be important, it is simply becoming a component of and not the sole technique for this line of research. This trend should be reflected, as well.

Third, please discuss marine examples of gnotobiotic marine species in "Emerging methodologies to approach the complexity of holobiont partnerships."


Minor:
Ln 75: Provide an alternate word to “stable” because transient interactions apply, as does the potential for symbioses to fade in or out after a couple generations
Ln 177: I am not sure that I would consider this an emerging concept per se. I'd say it is well-established, especially in the ocean (e.g., corals and sponges) but becoming more widely recognized amongst taxa. Please revise to reflect this.
Ln 239: Perhaps add Burgess et al. (2016, Biological Reviews) here too, as they provide a nice marine and terrestrial perspective on dispersal.
Ln 294: This comes as that Exaiptasia is a coral, which it is not. Please revise to reflect that it is a sea anemone.

Experimental design

See above.

Validity of the findings

See above

Additional comments

See above

Reviewer 2 ·

Basic reporting

This review is the result of a workshop in Roscoff Biological Station who brought together scientists working on marine holobionts from different disciplines and approaches. The review summarizes key concepts of holobiont research applied to the marine system. It discusses specific issues and opportunities related to the research of host-microbe interactions in the ocean. Finally, it emphasizes the need of a link to this research to management strategies. I find this review is valuable for multiple researchers in symbiosis research and marine science field. It is also a valuable resource for teaching! The Authors also mentioned policymakers as a target audience (L183-184). I think that is extremely valuable, but the vocabulary in the current version may not be plain enough or applied enough for that audience. One suggestion is to be more specific on the potential translations for management in the abstract. Why is the holobiont concept important for marine ecosystems and why now? My comments on the structure of the review are reported under Study design.

Experimental design

I much enjoyed the section “the history of the holobiont concept”. I only missed a couple of sentences at the beginning of the first paragraph for a strong statement about why it is the holistic perspective and the holobiont concept important now. I think of the policymakers here again. The paragraph and the whole section develop then this idea nicely, but a strong statement right at the start would help to engage them.
Section “marine holobiont models”. I miss more explanation on the advantages and disadvantages of “environmental” vs “tractable” models. Regarding the paragraph on environmental models, I also do not find well justified the mention of some systems and not others, nor the order they are presented. For example, the paragraph could be structured according to phylogeny, or according to autotrophic vs heterotrophic symbionts, or unicellular then multicellular. I am a bit surprised that sponges, corals and seaweeds are not mentioned or only one species is referred. I find that for environmental models the big taxonomic group can be mentioned, followed by one example of a particular species. Just an idea. The paragraph on tractable models is easier to follow.
Figure 1: I love how it shows all the diversity of marine holobionts, in particular the unicellular hosts, often overlooked in other reviews. But if your target audience includes policymakers, you may consider including a second panel in the figure with bigger picture of keystone holobionts or holobionts of particular interest for ecosystem functioning and services. This is just a suggestion, Authors must decide.
Section “marine holobiont assembly and regulation”. Here I see some problems with the flow of the section. First paragraph focuses on modes of transmission. Second paragraph starts referring to immunity as a mechanism for microbe selection during transmission, and then present dysbiosis as perturbation of microbial transmission and maintenance. There are too many concepts mixed here and not easy to follow. Also, the perturbation of the transmission not necessarily brings the same issues that problems during maintenance. Moreover, the Anna Karenina principle is an example of patterns related to non-deterministic processes and is in the middle of the selection paragraph without detailed explanation. My suggestion is to move the dysbiosis and “Russian dramas” (lines 362-375) to L397. Thus, the second paragraph of this section will focus on selection mechanisms: immunity and chemicals. The third paragraph will discuss other factors shaping the microbiome. The forth paragraph will bring these concepts to the health/disease context, and you end with the conclusive paragraph.
I suggest revising the definition of the Anna Karenina principle. The definition by Zaneveld et al. does not refer to number of parameter but rather to deterministic processes in health vs stochastic process in disease.

Validity of the findings

The conclusion puts together the perspectives on marine holobiont research.

Reviewer 3 ·

Basic reporting

no comment

Experimental design

no comment

Validity of the findings

no comment

Additional comments

Specific points:

Line 269: In this paragraph, I miss a statement to the requirements of marine holobiont model organisms. A table could be helpful summarizing valuable characteristics of mentioned marine holobionts, based on points summarized e.g. in Ruby EG 2008 Nat Rev Microbiol., Box 1.

Line 361/362: The selection of the microbiota by the immune system is an important aspect and should be elaborated. In addition, citations are missing for this claim. (e.g. Franzenburg et al 2013 PNAS). The subsequently mentioned concepts “Anna Karenina principle” and “Rasputin effect” are only of secondary importance for this point.

Line 452. Omics approaches are not the key for functional studies. Similar to large scale ecological studies –omics studies remain correlative and require experimental validation using e.g. CRISPR/cas9 genome editing on the host side or cultivation and transformation on bacterial side. This point needs a deeper discussion in this paragraph.

Line 568: The same is true here. Here, also functional analyses via transgensis (e.g. CRISPR/cas9) is missing.

Line 570: Not clear, what is meant here.

Line 574: What kind of “societal, and economic questions” are meant here? To my mind these kind of questions are not part of the review. Either include a chapter discussing these kind of questions or delete it here.

---

## Round 0.2 · accepted · Accept

Thank you for your thorough revision and patience during the reviewing process.

·

Basic reporting

I am happy with the revision by the authors and support publication of this manuscript.

Experimental design

I am happy with the revision by the authors and support publication of this manuscript.

Validity of the findings

I am happy with the revision by the authors and support publication of this manuscript.

Additional comments

I am happy with the revision by the authors and support publication of this manuscript.

Reviewer 2 ·

Basic reporting

no comment

Experimental design

no comment

Validity of the findings

no comment

Additional comments

The Authors have addressed my comments and those of the others reviewers.

Reviewer 3 ·

Basic reporting

no comment

Experimental design

no comment

Validity of the findings

no comment

Additional comments

I have no further comments.